# Generation of Herbicide-Resistant Soybean by Base Editing

**DOI:** 10.3390/biology12050741

**Published:** 2023-05-19

**Authors:** Tao Wei, Linjian Jiang, Xiang You, Pengyu Ma, Zhen Xi, Ning Ning Wang

**Affiliations:** 1Tianjin Key Laboratory of Protein Sciences, Department of Plant Biology and Ecology, College of Life Sciences, Nankai University, Tianjin 300071, China; weitao88888@163.com (T.W.); xiang.you@pku-iaas.edu.cn (X.Y.); 2State Key Laboratory of Elemento-Organic Chemistry, Department of Chemical Biology, National Engineering Research Center of Pesticide, College of Chemistry, Nankai University, Tianjin 300071, China; mapengyu0216@163.com; 3Key Laboratory of Pest Monitoring and Green Management, MOA, Department of Plant Pathology, College of Plant Protection, China Agricultural University, Beijing 100193, China; jianglinjian@cau.edu.cn

**Keywords:** soybean, *GmAHAS4*, herbicide-resistant, base editing

## Abstract

**Simple Summary:**

Soybean is a prominent grain and oil crop in the world. The production of soybean is threatened by weed competition. Recently, CRISPR/Cas9-mediated base editing, which can cause irreversible nucleotide changes in target loci, has been widely used in basic plant science and breeding. However, until now, herbicide-resistant traits in soybean have not been created by the base editor. The objective of this study was to modify different AHAS homologous alleles by base editor to generate a novel AHAS-inhibiting herbicide-resistant soybean. Our results showed that the GmAHAS4 P180S mutants have apparent resistance to AHAS-inhibiting herbicides, which will be beneficial for weed control in soybean production.

**Abstract:**

Weeds cause the largest yield loss in soybean production. The development of herbicide-resistant soybean germplasm is of great significance for weed control and yield improvement. In this study, we used the cytosine base editor (BE3) to develop novel herbicide-resistant soybean. We have successfully introduced base substitutions in *GmAHAS3* and *GmAHAS4* and obtained a heritable transgene-free soybean with homozygous P180S mutation in *GmAHAS4*. The GmAHAS4 P180S mutants have apparent resistance to chlorsulfuron, flucarbazone-sodium, and flumetsulam. In particular, the resistance to chlorsulfuron was more than 100 times that of with wild type TL-1. The agronomic performance of the GmAHAS4 P180S mutants showed no significant differences to TL-1 under natural growth conditions. In addition, we developed allele-specific PCR markers for the GmAHAS4 P180S mutants, which can easily discriminate homozygous, heterozygous mutants, and wild-type plants. This study demonstrates a feasible and effective way to generate herbicide-resistant soybean by using CRISPR/Cas9-mediated base editing.

## 1. Introduction

In fields, weeds not only compete with crops for light, growth space, and nutrients, but also spread pests and diseases and release toxic substances to reduce food production [1]; therefore, effective weed control is critical for world food security [2]. Currently, herbicides are extensively used to control weeds. However, improper use of herbicides can cause serious problems such as crop injury and the development of herbicide-resistant weeds [3]. Generating herbicide-resistant cultivars provides an effective solution [4]. On the one hand, herbicide-resistant varieties effectively reduced the risk of herbicide injury on crops; on the other hand, herbicide-resistant varieties with new mutations could introduce new herbicides to control recalcitrant weeds.

Soybean (*Glycine max* (L.) Merr.) is a prominent grain and oil crop in the world. Soybean is easily disturbed by weeds because the seeds are planted at a wide interval to form branches and to allow the canopy to expand fully in the later stage of growth. The late canopy closure makes soybean more prone to weeds than other crops [5]. Weeds cause the largest yield loss in soybean production. On a global basis, there is evidence that 37% of soybean yields is threatened by weed competition [6]. Therefore, creating herbicide resistant soybean germplasm is of great significance for weed control and yield improvement. However, the efficiency of using traditional breeding tools to create herbicide-resistant varieties is very low. Recently, a CRISPR/Cas9-mediated base editor which could efficiently and precisely perform C to T base editing in mammalian cells has been developed [7]. This tool was quickly applied to different plant species, generating several novel traits with significant agricultural values [8].

Acetohydroxy acid synthase (AHAS), a key enzyme for the biosynthesis of branched-chain amino acids including valine, leucine, and isoleucine, is the target of many herbicides with different chemical properties [9]. These types of herbicides, including imidazolinones (IMI), pyrimidinyl thiobenzoates (PTB), sulfonylaminocarbonyl triazolinones (SCT), sulfonylureas (SU), and triazolopyrimidines (TP), are widely used for weed control in crop fields, including soybean [10]. Thus, the creation of a herbicide-insensitive AHAS enzyme is of great value in agricultural application. Mutations in key amino acids of target enzymes can cause target site resistance, which is the most common mechanism leading to resistance to AHAS-inhibiting herbicides [11]. At present, in weed biotypes with field evolutionary resistance, at least 28 amino acid substitutions conferring resistance have been identified in eight conserved positions of AHAS enzymes [12]. Among them, the most common mutation site is the P197 codon (numbered according to the corresponding sequence of *A. thaliana*) [13]. Recently, the P197 site has been modified by the base editor to generate herbicide-resistant germplasm in many crops [4]. However, until now, herbicide-resistant traits have not been created by base editor in soybean.

In soybean, four homologs of *AHAS* have been identified, including *GmAHAS1* (Glyma04g37270), *GmAHAS2* (Glyma06g17790), *GmAHAS3* (Glyma13g31470), and *GmAHAS4* (Glyma15g07860). C to T mutations were once successfully introduced to the P178 codon of *GmAHAS1* via chemical mutagenesis [14], which conferred a moderate resistant level to sulfonylurea herbicides. However, it is unknown whether mutations at the P197 site (numbered according to the corresponding sequence of *A. thaliana*) of other homologous alleles will endow soybean resistance to herbicides. The objective of the present study was to modify different AHAS homologous alleles by base editing technology to create a novel herbicide-resistant soybean.

## 2. Materials and Methods

### 2.1. Vector Construction and Soybean Transformation

Close analysis revealed that the sequences of *GmAHAS3* and *GmAHAS4* in the P197 region are identical, and the sequence of *GmAHAS2* is different from that of *GmAHAS3* and *GmAHAS4* with only one base ((Figure 1), so the P197 region of *GmAHAS2* and *GmAHAS3/4* were selected as the target sites for creating herbicide-resistant soybean by using a base editing system. These two selected target sequences (AGGTCCCCCGGCGCATGAT and AGGTCCCCCGCCGCATGAT) were cloned into pBSE901, respectively [15], in which sgRNA transcription was driven by the *Arabidopsis* U6 promoter and BE3 was driven by double 35S promoter. The *bar* gene, encoding phosphinothricin acetyltransferase (PAT) and conferring tolerance to phosphinothricin (PPT), was used as a selectable marker. These two binary vectors were then transformed into *Glycine max* (L.) cultivar Tianlong 1 (TL-1), as described previously [16].

### 2.2. Mutation and Transgene-Free Detection

For mutation detection, genomic DNA from the leaf tissue of T0 transgenic plants and their progenies was extracted using a modified CTAB method [17]. The target regions of *GmAHAS1-4* were amplified by PCR using gene-specific primers and the products were subjected to Sanger sequencing.

To identify transgene-free plants, the gene-specific primer pairs U6-26p-F/U6-26t-R, Cas9-F/Cas9-R, and Bar-F/Bar-R were used to amplify the sgRNA, Cas9, and Bar genes, respectively. Plants-lacking all three genes concurrently, were considered to be free of transgenes.

### 2.3. Off-Target Detection

Potential off-target sites, which contained 1–5 nucleotide mismatches relative to the *GmAHAS3/4* target sites, were identified by CRISPR-GE “http://skl.scau.edu.cn/targetdesign/ (accessed on 18 October 2020)” [18]. The potential off-target sites were amplified with the gene-specific-primers (Appendix A) and detected by Sanger sequencing.

### 2.4. Herbicide Tolerance Assay

Soybean seedlings were cultured in a greenhouse with 14 h light at 250 to 300 μmol photons/m^2^/s and 10 h dark at 25 °C with 60% humidity. Five representative herbicides from five AHAS inhibitors families, including bispyribac-sodium, imazapyr, chlorsulfuron, flucarbazone-sodium, and flumetsulam, were sprayed once at the V2 stage. The concentration gradient of each herbicide was 0, 1, 10, 100, and 1000 mg/L, respectively. Phenotypes and plant height of GmAHAS4 P180S mutants and wild-type plants were recorded 10 days after herbicide treatment. Each treatment was repeated at least three times.

### 2.5. Development of Allele-Specific PCR Markers

To distinguish GmAHAS4 P180S mutant from WT plants, two pairs of primers were designed. A common forward primer (GmAHAS4-P180-F) was *GmAHAS4*-specific and could differentiate between *GmAHAS4* and other homologous genes. The two base editing sites were chosen to design at the 3′ end of the reverse primers. The wild type *GmAHAS4* gene was amplified by GmAHAS4-P180-F/GmAHAS4-180P-R, and the P180S mutant *GmAHAS4* gene was amplified by GmAHAS4-P180-F/GmAHAS4-180S-R.

## 3. Results

### 3.1. Targeted Base Editing of GmAHAS Genes

A total of 8 T0 transgenic plants with *GmAHAS2*-targeting vector and 15 T0 transgenic plants with *GmAHAS3/4*-targeting vector were obtained after screening by the PAT testing strips. Sanger sequencing results revealed that no gene editing occurred in 8 transgenic plants with *GmAHAS2*-targeting vector, 5 out of 15 T0 transgenic plants with *GmAHAS3/4*-targeting vector exhibited editing events, among which 3 plants (Line 1, 5, and 11) were edited at the target sites of *GmAHAS3* and *GmAHAS4* simultaneously, 1 plant (Line 10) only mutated at the target site of *GmAHAS3*, and the other plant (Line 8) only mutated at the target site of *GmAHAS4* (Figure 2). Interestingly, most of these mutations are C to T, and in Line10, in addition to C to T, we also found C to A mutation. Furthermore, we found that these plants are all chimeric mutants, and the mutation levels at different sites in different plants are different. To assess the possible off-target events of the five base editing mutations, a total of nine putative off-target sites (including *GmAHAS1* and *GmAHAS2*) were predicted by CRISPR-GE. After sequencing the PCR products of the potential off-target sites, we did not detect any off-target events in these gene-edited plants (Appendix A).

### 3.2. Identification of GmAHAS4 P180S Mutants

To test whether the base-edited mutations could be passed to the next generation efficiently, and to obtain heritable homozygous mutants, T1 progeny of five T0-edited plants were analyzed. Based on the sequencing results, only the mutation of Line 8 was stably inherited to the T1 generation, and a total of 23 wild-type plants (P180P/P180P), 18 heterozygous mutants (P180P/P180S), and 14 homozygous mutants (P180S/P180S) were obtained (Figure 3A). PCR amplification of *Cas9*, *sgRNA*, and *Bar* genes showed that a total of 15 T1 mutants lacked the exogenous T-DNA due to genetic segregation (Figure 3B). To examine the stable transmission of P180S allele, T1 homozygous P180S mutants without transgenic components were cultured to T3 generation. Sequencing results showed that these T3 plants all contained homozygous P180S mutations (Appendix A), and PCR amplification results showed that they did not contain transgenic components (Appendix A). Meanwhile, no new editing events were detected in the target region of GmAHAS1-3 and seven putative off-target sites of the T3 P180S mutants (Appendix A).

### 3.3. Herbicide Tolerance of GmAHAS4 P180S Mutants

To determine whether the base editing mutants could endow soybean with herbicide resistance, the wild-type controls (TL-1), together with the transgene-free T3 homozygous GmAHAS4 P180S plants, were treated with different concentrations of bispyribac-sodium, imazapyr, chlorsulfuron, flucarbazone-sodium, and flumetsulam at the V2 stage. As shown in Figure 4, there was no phenotypic difference between the GmAHAS4 P180S mutants and the wild-type controls (TL-1) without herbicide treatment. Compared with the wild-type TL-1, the GmAHAS4 P180S mutants showed slight resistance to bispyribac-sodium and no apparent resistance to imazapyr. As for chlorsulfuron, the wild-type TL-1 cannot grow at a concentration of 1 mg/L, while the GmAHAS4 P180S mutant can still maintain growth at 100 mg/L. As for flucarbazone–sodium, the wild-type plants died quickly at 100 mg/L, while the mutant survived at 1000 mg/L, although its growth was severely inhibited. As for flumetsulam, both TL-1 and mutants could maintain growth at the concentration of 100 mg/L. When the concentration increased to 1000 mg/L, the wild-type TL-1 plants died rapidly, while the GmAHAS4 P180S mutant maintained growth.

### 3.4. Agronomic Traits of GmAHAS4 P180S Mutants

To evaluate the effect of the P180S allele on agronomic traits, the transgene-free T3 homozygous GmAHAS4 P180S soybean were grown in the greenhouse under natural growth conditions in Tianjin, China. As shown in Figure 5, the agronomic traits, including the morphology of plant and seed, seed number per plant, seed yield per plant, and hundred-grain weight, showed no significant differences between TL-1 and the P180S plants.

Furthermore, we designed allele-specific PCR markers to distinguish GmAHAS4 P180S mutants from the wild-type TL-1 plants. The upstream primer was designed according to the specific sequence of *GmAHAS4*, and two base editing sites were designed to the 3’ end of the downstream primer (Figure 6A). The results showed that these primers could easily discriminate homozygous mutants, heterozygous mutants and wild-type plants (Figure 6B).

## 4. Discussion

Due to its simplicity, specificity, efficiency, and versatility, the CRISPR/Cas system has quickly become the preferred tool for plant genome engineering [19,20,21]. The application of the CRISPR/Cas system in plant genome editing mainly includes gene knockout, gene targeted insertion or replacement, and gene expression regulation [22]. Recently, as a new gene-editing strategy, CRISPR/Cas-mediated base editing, which can cause irreversible nucleotide changes in target loci without double stranded DNA cleavage or any donor template, has been widely used in basic plant science and breeding [23,24]. For example, C to G substitution with frequency of 1.6% to 3.9% and C to T substitution with frequency of 1.4% to 11.5% were detected in rice by using base editing. In addition, C to T substitution in SLR1 (S97L) resulted in a distinct semi dwarf phenotype [25]. To facilitate weed control, herbicide-resistant germplasm has been generated via the base editor in many crops, including rice [26], wheat [2], canola [27], tomato [26], and watermelon [15]. In the present study, we used the cytosine base editor BE3 to generate a novel AHAS-inhibiting herbicide-resistant soybean mutant. Our results showed that the GmAHAS4 P180S mutants have apparent resistance to chlorsulfuron, flucarbazone-sodium, and flumetsulam.

Since different types of AHAS-inhibiting herbicides bind to various sites of the herbicide binding domain, particular point mutation of the AHAS gene could confer tolerance to a specific range of AHAS-inhibiting herbicides [28]. Meanwhile, the herbicide resistance conferred by AHAS mutation has a dose effect [29]. In rice, the four different mutations in the P197 codon—P197A, P197S, P197Y, and P197F—exhibited different resistance patterns towards different AHAS-inhibiting herbicides. P197S and P197A showed themselves to be more sensitive to bispyribac than P197Y and P197F [28]. In wheat, the P197S conferred higher resistance to nicosulfuron than P197A [2]. In this study, the GmAHAS4 P180S mutants showed no apparent resistance to imazapyr, slight resistance to bispyribac–sodium, and apparent resistance to flumetsulam. The resistance to flucarbazone-sodium was more than 10 times and the resistance to chlorsulfuron was more than 100 times.

Gene editing efficiency and inheritance of genome modifications are two important considerations when using CRISPR/Cas9-mediated gene editing technology for basic research and molecular breeding [16]. Michno et al. reported that the integration and inheritance of CRISPR/Cas9 soybean lines include three patterns, nanemly, no transmission of mutations and transgenes, inheritance of transgenes located within the target sites but no mutations, and transmission of mutations but segregation of transgenes [30]. In CRISPR/Cas9-induced mutations, the loss of mutations and transgenes in subsequent generations seems to be common. Shi et al. reported that neither transgene nor mutations of two T0 generation soybean PDS gene editing lines were inherited by T1 progenies [17]. They found that the indel frequencies of the T0 generation were low in both strains, so the T0 plants were likely chimeric, and the mutation only existed in somatic cells, so it was not inherited into the T1 generation. In the present study, we did not obtain *GmAHAS2* gene editing mutants in T0 transgenic plants. Meanwhile, although five *GmAHAS3/4* gene editing mutants were obtained in the T0 transgenic plants, only one mutation was stably inherited to the next generation. The possible reasons for these phenomena mainly include the following aspects: (i) the genetic transformation efficiency of soybean was low [31], and the obtained transgenic plants were too few; (ii) the vector used in this study has low editing efficiency in soybean [32]; (iii) the T0 transgenic regenerated plants obtained by tissue culture were chimeric, and the mutations were only present in somatic cells [17]; thus, they cannot be inherited to the next generation. Therefore, systematic optimization of the soybean transgenic regeneration system and the gene-editing system to reduce the generation of chimeras and false-positive plants is of great significance for improving the efficiency of heritable gene editing mutation in soybean.

## 5. Conclusions

In summary, we have successfully introduced base substitutions in *GmAHAS3* and *GmAHAS4* by CRISPR/Cas-mediated base-editing and obtained a transgene-free herbicide-resistant soybean with homozygous P180S mutation in *GmAHAS4*. This P180S mutation has no adverse effects on agronomic traits. Moreover, we successfully designed allele-specific markers to identify P180S mutations of *GmAHAS4*, which will help breed herbicide-resistant soybean. In the future, multi-point mutation of the AHAS gene and simultaneous modification of different homologs will be beneficial to obtain soybean with multiple herbicide resistance, thus contributing to weed management in soybean production.

## Figures and Tables

**Figure 1 biology-12-00741-f001:**
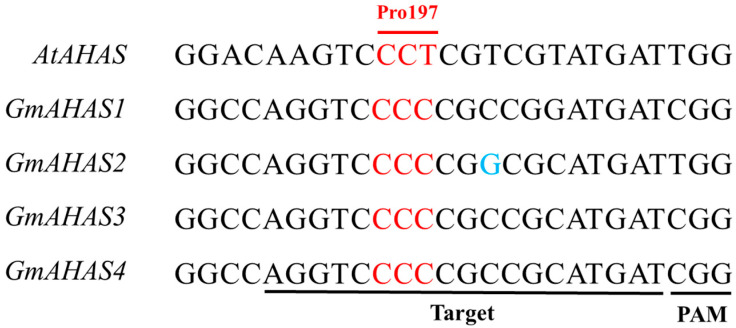
The alignment of target regions of *AtAHAS* and *GmAHAS* genes. The P197 of *AtAHAS* are listed, and corresponding codons of *GmAHAS* genes are shown in red. The target sites of *GmAHAS* genes were marked with underscores.

**Figure 2 biology-12-00741-f002:**
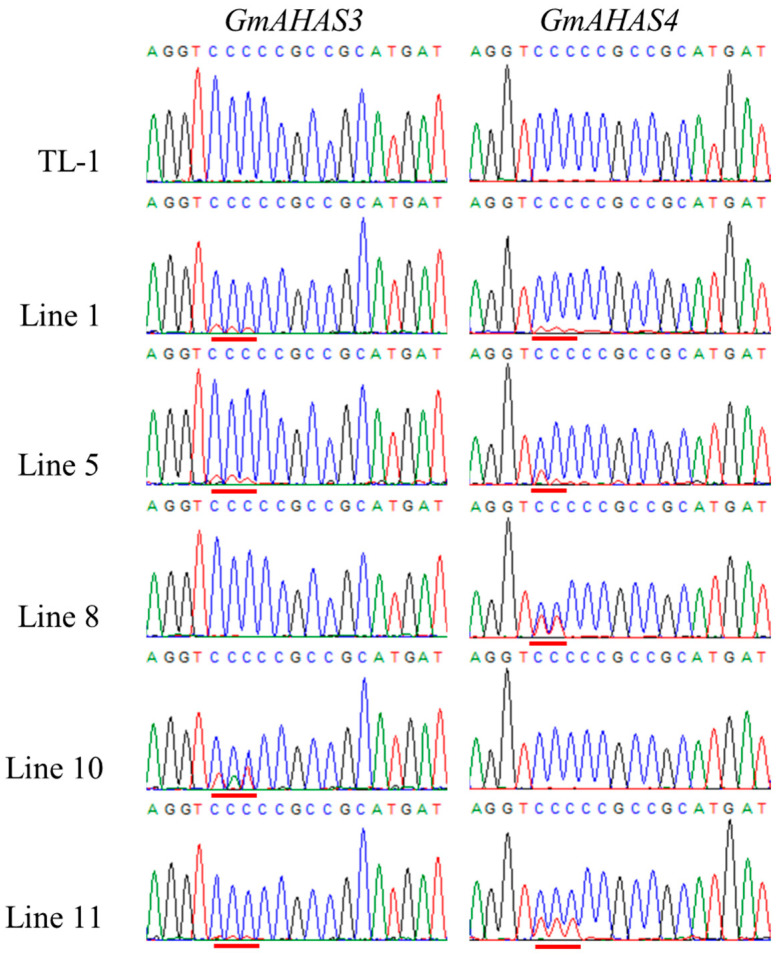
Base editing of *GmAHAS* genes. Chromatograms of Sanger sequencing results of 5 T0 plants with editing events. Red lines indicate mutation locations.

**Figure 3 biology-12-00741-f003:**
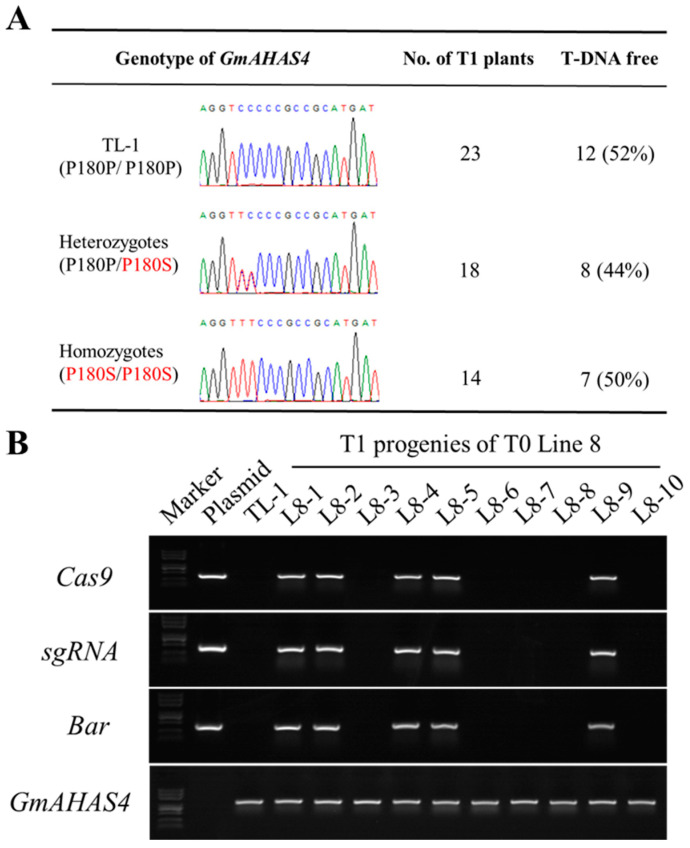
Identification of transgene-free GmAHAS4 P180S homozygous mutants. (**A**) The mutation types of GmAHAS4 in the T1 generation of line 8 according to the Sanger sequencing results. (**B**) Isolation of T1 plants without T-DNA insert. The presence and absence of individual genes were detected by PCR-amplification with gene-specific primers (Cas9, sgRNA, Bar).

**Figure 4 biology-12-00741-f004:**
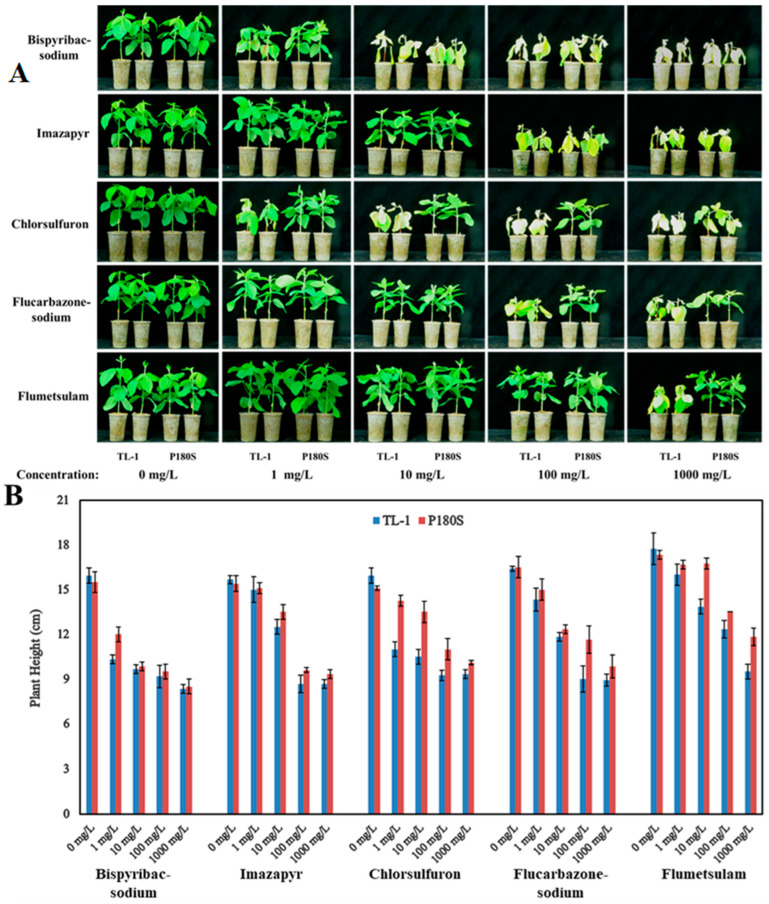
Resistance of GmAHAS4 P180S to five AHAS-inhibiting herbicides. (**A**) Phenotype of GmAHAS4 P180S after herbicides treatment for 10 days. (**B**) Plant height of GmAHAS4 P180S after herbicides treatment for 10 days.

**Figure 5 biology-12-00741-f005:**
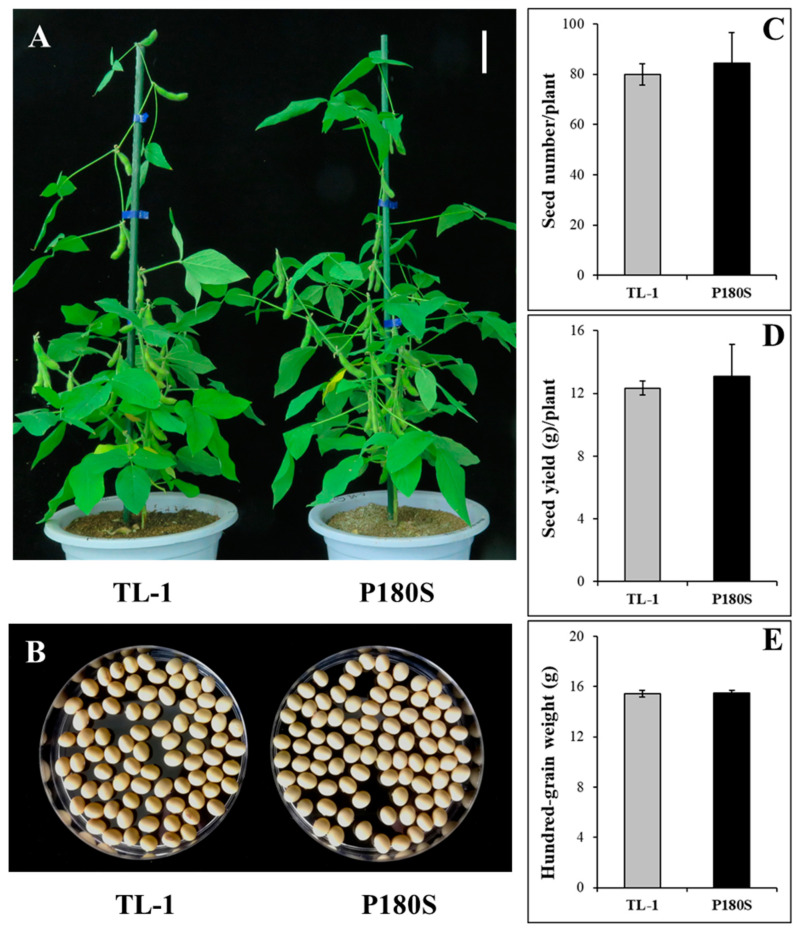
Agronomic traits of GmAHAS4 P180S mutants. (**A**) Morphology of plant (bar = 10 cm). (**B**) Morphology of seed. (**C**) Seed number per plant. (**D**) Seed yield per plant. (**E**) Hundred-grain weight.

**Figure 6 biology-12-00741-f006:**
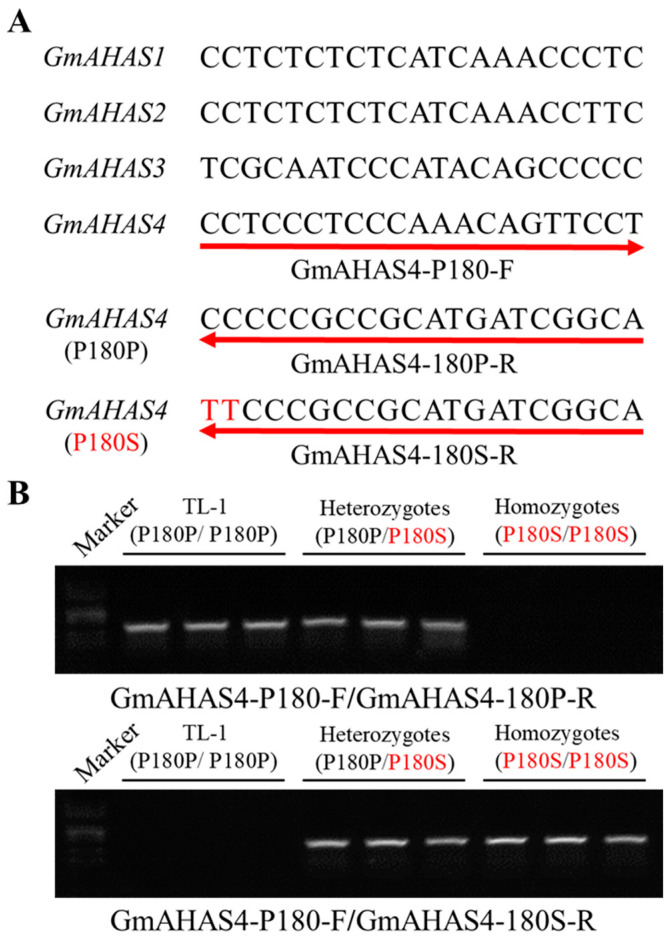
Allele-specific primer design (**A**) and PCR amplification of GmAHAS4 P180S homozygous, heterozygous mutants and wild type plants (**B**).

## Data Availability

All data used to support the findings of this study are included within the article and they are also available from the corresponding author upon request.

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
