# Peer review of "Generation of Herbicide-Resistant Soybean by Base Editing"

_biology, 2023, doi:10.3390/biology12050741_

Round 1

Reviewer 1 Report

Introduction

The authors should highlight the importance of acetohydroxyacid synthase (AHAS) gene. Please explain why this gene was targeted.

The Pro197 site has been modified by CRISPR/CAS in other crops. These should be mentioned in introduction and results should be compared in discussion.

Materials and methods:

Provide specific information on

i) guide sequence targeting the genes were designed

ii) Confirm whether you have used PBSE901 or PHEE901 as vector, as the reference the authors have cited used PHEE901. Also cite the source of the vector

iii)  Use a standard reference for herbicide tolerance test assay, or if the assay is designed by the authors it should be mentioned with reason

Results

Line 118-119: Explain chimeric mutants. For which character chimera was observed?

Line 122-23: if the editing events were not responsible for chimera as predicted, what may be possible cause?

Line 134: Some some T1 mutants lacked the exogenous T-DNA--- how many? Mention in text

Line 150: What was the basis fro selection of the herbicides?

Discussion

The first paragraph of the discussion should be shifted to introduction

Line 204-207 discussions are speculations, not based on result.

Line 209-211 discussions are not based on result

The discussion section needs major improvement. the authors have not mentioned the gene editing effects in the plants, the segretaion of non-transgenic to transgenic plants, the explanations for difference in herbicide treatment, why the edited plant is not effective against some herbicide but highly effective against others etc. are not discussed. 

Improve the style of language. Grammar is OK

Author Response

  1. The authors should highlight the importance of acetohydroxyacid synthase (AHAS) gene. Please explain why this gene was targeted.

Answer: Thanks for the suggestion. We have revised the introduction of the manuscript to highlight the importance of the AHAS gene. In lines 63-73, we have added the following sentences in the Introduction section of the revised manuscript:

Acetohydroxy acid synthase (AHAS), a key enzyme for the biosynthesis of branched-chain amino acids including valine, leucine and isoleucine, is the target of many herbicides with different chemical properties [9]. These types of herbicides, including imidazolinones (IMI), pyrimidinyl thiobenzoates (PTB), sulfonylaminocarbonyl triazolinones (SCT), sulfonylureas (SU), and triazolopyrimidines (TP), are widely used for weed control in crop fields, including soybean [10]. Thus, the creation of herbicide-insensitive AHAS enzyme is of great value in agricultural application. Mutations in key amino acids of target enzymes can cause target-site resistance, which is the most common mechanism leading to resistance to AHAS-inhibiting herbicides [11]. At present, in weed biotypes with field evolutionary resistance, at least 28 amino acid substitutions conferring resistance have been identified in eight conserved positions of AHAS enzymes [12].

  1. The Pro197 site has been modified by CRISPR/CAS in other crops. These should be mentioned in introduction and results should be compared in discussion.

Answer: Thanks for the suggestion. We mentioned Pro197 site in the introduction of the revised manuscript (see lines 74-77) and compared it in the discussion section (see lines 246-256).

Materials and methods:

Provide specific information on

  1. i) guide sequence targeting the genes were designed

Answer: Thanks for the suggestion. We have provided the guide sequence in the revised manuscript (see line 96).

  1. ii) Confirm whether you have used PBSE901 or PHEE901 as vector, as the reference the authors have cited used PHEE901. Also cite the source of the vector

Answer: Thanks for the notification. We used PBSE901as vector in our experiment, and we made corrections to the citation in the revised manuscript (see line 97).

  1. iii) Use a standard reference for herbicide tolerance test assay, or if the assay is designed by the authors it should be mentioned with reason

Answer: Thanks for the suggestion. Due to the lack of relevant reference standards, we designed herbicide tolerance test assay through preliminary exploration.

Results

  1. Line 118-119: Explain chimeric mutants. For which character chimera was observed?

Answer: Thanks for the question. Chimeric mutants refer to individuals in which a portion of cells carry mutations while another portion of cells do not. From the peak plot of the sequencing results, we can observe chimeric mutations, while the relevant characters are only the results read by the sequencing machine.

  1. Line 122-23: if the editing events were not responsible for chimera as predicted, what may be possible cause?

Answer: Thanks for the question. The reason for this phenomenon may be that base mismatch occurs in DNA replication.

  1. Line 134: Some some T1 mutants lacked the exogenous T-DNA--- how many? Mention in text

Answer: Thanks for the suggestion. We have supplemented related content in the revised manuscript (see lines 162-163).

  1. Line 150: What was the basis fro selection of the herbicides?

Answer: Thanks for the question. AHAS-inhibiting herbicides are mainly divided into five categories, and we selected representatives from each category for experiments.

Discussion

  1. The first paragraph of the discussion should be shifted to introduction

Answer: Thanks for the suggestion. We have shifted the first paragraph of the discussion to introduction (see lines 63-77).

  1. Line 204-207 discussions are speculations, not based on result.

Answer: Thanks for the comments. Based on the experimental results and referring to reports from others, we have made reasonable speculations about the cause.

  1. Line 209-211 discussions are not based on result

Answer: Thanks for the comment. Based on the experimental results and referring to reports from others, we have made reasonable speculations about the cause.

  1. The discussion section needs major improvement. the authors have not mentioned the gene editing effects in the plants, the segretaion of non-transgenic to transgenic plants, the explanations for difference in herbicide treatment, why the edited plant is not effective against some herbicide but highly effective against others etc. are not discussed. 

Answer: Thanks for the suggestion. We have conducted a major improvement of the discussion section and added relevant content (see lines 230-280).

Reviewer 2 Report

The introduction has sufficient and relevant information.

The methodology is sound and clearly presented.

The results are straightforward and adequately analyzed.

The conclusion is based on the results.

The discussion section could be enriched with more examples.

Author Response

  1. The introduction has sufficient and relevant information.

Answer: Thanks for the comments.

  1. The methodology is sound and clearly presented.

Answer: Thanks for the comments.

  1. The results are straightforward and adequately analyzed.

Answer: Thanks for the comments.

  1. The conclusion is based on the results.

Answer: Thanks for the comments.

  1. The discussion section could be enriched with more examples.

Answer: Thanks for the suggestion. We have conducted a major improvement of the discussion section and added relevant content (see lines 230-280).

Reviewer 3 Report

Wei et al in their manuscript entitled “Generation of herbicide-resistant soybean by base-editing” have developed novel herbicide-resistant soybean by a CRISPR/Cas9-assisted C to T base editing. The authors were able to generate heritable homozygous P180S mutation in GmAHAS4, with minimal impact on normal growth of the plant in general. Primers distinguishing genotypic variations in different generations are also reported. Such work is of significance and would be of value to the readers of Biology.

There remain a few minor concerns which could be addressed by the authors, which are as follows: 

1.     In paragraph 4 of the introduction, author can add a few lines on AHAS being preferred herbicidal resistance gene for more clarity.

2.     In line 86, “Plants, which lacking” should be changed to “Plants lacking”.

3.     Line 93 can say either test or assay, it seems repetitive.

4.     In figure 4B, the lines in the background can be got rid of. X axis can be made simpler by mentioning the name of herbicide collectively at the bottom of each set.

5.     In general, same font and size should be used in all the figures to make things look even.

Author Response

  1. In paragraph 4 of the introduction, author can add a few lines on AHAS being preferred herbicidal resistance gene for more clarity.

Answer: Thanks for the suggestion. We have revised the introduction of the manuscript to highlight the importance of the AHAS gene.

  1. In line 86, “Plants, which lacking” should be changed to “Plants lacking”.

Answer: Thank you for your correction. We have changed “Plants, which lacking” to “Plants lacking” (see line 113).

  1. Line 93 can say either test or assay, it seems repetitive.

Answer: Thank you for your correction. We have changed “Herbicide tolerance test assay” to “Herbicide tolerance assay” (see line 120).

  1. In figure 4B, the lines in the background can be got rid of. X axis can be made simpler by mentioning the name of herbicide collectively at the bottom of each set.

Answer: Thanks for the suggestion. We have made modifications to Figure 4B (see line 191).

  1. In general, same font and size should be used in all the figures to make things look even.

Answer: Thanks for the suggestion. We have made modifications to the relevant figures.